# CHALLENGE REPORT

## ABSTRACT

This brief report describes an approach to modify the graph inference function for the causal bench challenge which is based on the dataset described in Chevalley et al. (2022).

## 1 METHOD OVERVIEW

We observe that for the DCDI algorithm the genes are partitioned in small groups and the algorithm is applied to those groups independently. Even for rather small number of genes, 500 say, and 50 nodes per partition element, the probability of a specific edge to be included in one partition is only 10%. Thus a good choice of partitions can greatly increase the number of suitable candidate edges the DCDI algorithm can potentially find. Thus we constructed clusterings based on similarities of genes which might indicate closeness in the causal structure and therefore potentially graph edges. We then ran DCDI on the individual clusters.

We consider two suggestions to obtain the clustering. First we defined
$$d(k,l) = 1 - \text{corrcoef}(X_k, X_l) \tag{1}$$
where $X_k$ and $X_k$ are the expression of gene $k$ and $l$. Then we used spectral clustering using $d$. We fixed the average cluster size $n_{\text{avg}}$ and the maximal cluster size $n_{\text{max}}$ and split too large clusters randomly in two subclusters. In addition we considered the mean shifts between environments by
$$\mu_k^{(i)} = \mathbb{E}^{(i)}(X_k) - \mathbb{E}(X_k) \tag{2}$$
where $i$ denotes interventional distribution $i$. For each gene $k$ we thresholded
$$s_k^{(i)} = \begin{cases} 1 & \text{if } |\mu_k^{(i)}| \text{ is larger than 90\% of the } |\mu_k^{(j)}|, \\ 0 & \text{else.} \end{cases} \tag{3}$$
We define the similarity matrix $S_{kl} = \sum_i s_k^{(i)} s_l^{(i)}$. Then we construct partitions by randomly selecting a cluster seed (a randomly selected gene $k_1$) and set $C^1 = \{k_1\}$ and then greedily adding nodes $k_{n+1}$ to the cluster $C^n$ such that $k_{n+1} \in \text{argmax}_l \sum_{k_i \in C^n} S_{k_i l}$ until a fixed cluster size $n_{avg}$ is reached. This is repeated for 3 times. Again, the rationale is that genes whose expression levels change in a similar pattern for the provided interventional data might be close in the causal graph.

For the total of 4 partitions we run the DCDI algorithm and then threshold the edge probabilities for each run by
$$p \to \text{RELU}(p - .5), \tag{4}$$
i.e., we keep the information about edges with probability at least .5 predicted by the algorithm. In the end we add up the thresholded edge probabilities over the four partitions (this favours edges that end up in the same cluster for several partitions which is intended). The 2000 edges with the highest aggregated probabilities are returned by the algorithm. We chose $n_{\text{avg}} = 30$ as the average size or fixed size for the clusters and $n_{\text{max}} = 50$ as the maximal size after the script crashed for larger partition sizes.

All other parts of the inference function remained the same as in the provided code.

## REFERENCES

Mathieu Chevalley, Yusuf Roohani, Arash Mehrjou, Jure Leskovec, and Patrick Schwab. Causal-bench: A large-scale benchmark for network inference from single-cell perturbation data. *arXiv preprint arXiv:2210.17283*, 2022.

