# OpenReview forum: "Challenge Report"
_GSK.ai/2023/CBC_

### Official Review · Reviewer_QDPA · 2023-04-28
**Challenge Report - better partitions for DCDI algorithm**

**Rating:** 6
**Confidence:** 4

**Review:**

The authors likely noted that the DCDI algorithm is the best, or nearly the best, performing amongst the other benchmarks in CausalBench and enjoys some desirable properties, namely it improves as the dataset size grows or the number of interventions included grows.  However since it is implemented by first partitioning the set of genes at random into smaller, more tractable chunks, the authors propose to replace the random partition with one that accounts for gene similarity, with the hope that true edges of the graph will have a better chance of being edge candidates within the respective partition elements.

Pros:
1.  Aims to make one of the leading algorithms better by leveraging domain specific knowledge available in the data type.
2.  Uses both a global, correlation based metric and one based on the magnitude of average expression change under the interventional datasets to define the similarity matrix for spectral clustering.
3.  Some robustness to the random seed of the starting cluster

Cons:
1. Not a new algorithm, just modifies an existing approach and thus suffers from the same lack of scalability as DCDI
2. The similarity matrix  based on a gene's shift in mean expression under interventions considers only the difference in expected values.  That absolute magnitude comparison is probably not as biologically relevant as say a relative or percent difference

My take is that while this approach doesn't get much points for originality, given that the DCDI algorithms are intractable at the 'natural' scale of transcriptomics data and need to be applied piece-meal anyway, this is a reasonable addition to their application